# Divergence together with microbes: A comparative study of the associated microbiomes in the closely related *Littorina* species

**Arina L. Maltseva**[1], **Marina A. Varfolomeeva**[1], **Elizaveta R. Gafarova**[1], **Marina A. Z. Panova**[1,2]*, **Natalia A. Mikhailova**[3], **Andrei I. Granovitch**[1]

1 Department of Invertebrate Zoology, St Petersburg State University, St Petersburg, Russia, 2 Department of Marine Sciences -Tjärnö, University of Gothenburg, Gothenburg, Sweden, 3 Centre of Cell Technologies, Institute of Cytology Russian Academy of Sciences, St Petersburg, Russia

* marina.panova@marine.gu.se

**Data Availability Statement:** All sequence data are deposited in the NCBI archive, accession number PRJNA778532.

## Abstract

Any multicellular organism during its life is involved in relatively stable interactions with microorganisms. The organism and its microbiome make up a holobiont, possessing a unique set of characteristics and evolving as a whole system. This study aimed to evaluate the degree of the conservativeness of microbiomes associated with intertidal gastropods. We studied the composition and the geographic and phylogenetic variability of the gut and body surface microbiomes of five closely related sympatric *Littorina* (*Neritrema*) spp. and a more distant species, *L. littorea*, from the sister subgenus *Littorina* (*Littorina*). Although snail-associated microbiomes included many lineages (207–603), they were dominated by a small number of OTUs of the genera *Psychromonas*, *Vibrio*, and *Psychrilyobacter*. The geographic variability was greater than the interspecific differences at the same collection site. While the microbiomes of the six *Littorina* spp. did not differ at the high taxonomic level, the OTU composition differed between groups of cryptic species and subgenera. A few species-specific OTUs were detected within the collection sites; notably, such OTUs never dominated microbiomes. We conclude that the composition of the high-rank taxa of the associated microbiome ("scaffolding enterotype") is more evolutionarily conserved than the composition of the low-rank individual OTUs, which may be site- and / or species-specific.

## Introduction

Evolutionary biology is currently changing its focus from the evolution of an individual genome to that of a multigenomic system, the hologenome [1–8]. This shift of focus was prompted by the discovery of multiple and indispensable roles of microorganisms in the functioning of multicellular organisms, concerning aspects as diverse as metabolism, development, immunity, feeding and reproductive behaviour [9–25]. A profound impact of microorganisms on their multicellular hosts is associated with their inalienable contact throughout the

**Funding:** Financial support for this study was provided by Russian Scientific Foundation, grant 19-14-00321 to A.I. Granovitch (2019-2021). The visit to Tjärnö Marine Laboratory Research Station in 2018 was financed by the Royal Swedish Academy of Sciences, grant 3157/253144505 to A. L. Maltseva. The Swedish Academy of Science grant BS2018-0086 to M.A.Z. Panova and the Centre for Marine Evolutionary Biology CeMEB at University of Gothenburg covered sequencing costs. The funders did not play any role in the study design, data collection and analysis, decision to publish or preparation of the manuscript.

**Competing interests:** The authors have declared that no competing interests exist.

ontogenesis of an individual and the evolution of the species, and results in the genomic rearrangements in all participants [3, 4, 26, 27].

High-throughput sequencing methods have paved the way for a comprehensive microbiome research through metabarcoding and metagenomics. Nevertheless, our knowledge of the evolution of hologenomes and the role of microbes in the evolution of their hosts is mainly based on information obtained on mammals [10, 11, 20, 28] and terrestrial insects [13, 14, 17, 24, 25]. Among marine invertebrates (whose diversity makes more than 90% of aquatic species), associated microbiomes have been described for some sponges [29, 30], cnidarians [31, 32], and polychaetes [33, 34], but in general marine invertebrate remain poorly studied in this respect.

The molluscs are one of the largest invertebrate phyla, with the class Gastropoda being the richest in species. Littorinidae (Caenogastropoda) is one of the most thoroughly investigated gastropod families, and the genus *Littorina* has become a popular model for ecological speciation studies [35–37]. Although *Littorina* snails have been studied in many evolutionary aspects, including comparative genomics, ecology, physiology, biochemistry and parasitology [38–50], the description of their associated microbiomes is still in its infancy. Only the Pacific species *L. keenae* has been examined in this respect [51].

The Northern Atlantic *Littorina* spp. are distributed across two sister subgenera: *Littorina* subgen. *Littorina* Férussac, 1822 and *Littorina* subgen. *Neritrema* Récluz, 1869. The Atlantic *Neritrema* is a monophyletic clade, which includes two groups of closely related species: the "obtusata" group (*L. obtusata* [Linnaeus, 1758] and *L. fabalis* [W. Turton, 1825]) and the "saxatilis" group (*L. saxatilis* [Olivi, 1792], *L. arcana* Hannaford Ellis, 1978 and *L. compressa* Jeffreys, 1865). The subgenus *Littorina* is represented on the Atlantic shores only by *L. littorea* (Linnaeus, 1758). These six Atlantic species often coexist in sympatry at the intertidal zone though each species demonstrates a typical microdistribution pattern and microbiotope preferences [42, 49, 52]. Niche differentiation was proposed to be a key factor of the *Neritrema* species divergence, which probably occurred recently within the two cryptic groups (~0.5–1 Mya; [49, 50, 52, 53]). These six species represent a hierarchical system, with phylogenetic relationships of different depth and ecological specialization within the intertidal zone. This gives us ample opportunity to evaluate the impacts of niche differentiation and phylogeny on the associated microbiomes of these molluscs.

In this study, we described microbiomes associated with the six North Atlantic *Littorina* spp. We aimed to answer the following questions: (1) How strictly do *Littorina* snails control their associated microbiomes; that is, how do their microbiomes differ from the environment and how do they vary between individuals? (2) Do the snails have different microbiomes on the body surface and in the gut? (3) Are there differences in the associated microbiome composition between the species? (4) Does variation of the host-associated microbiomes correlate with the phylogenetic divergence of the hosts? These results lay ground for further studies of the functional role of associated microorganisms in the host evolution in *Littorina* in particular, and in marine invertebrates in general.

## Material and methods

### Sample collection

The sampling was performed in two regions: the Norwegian Sea, Tromsø, Norway and the North Sea, Tjärnö, Sweden. There were two sampling sites in each region (Tromsø#1 69˚ 38'11"N 18˚54'13"E; Tromsø#2 69 ˚43'41"N 18˚ 57'39"E; Tjärnö#1 58˚49'52.6"N 11˚04'00.8"E; Tjärnö#2 58˚49'58.1"N 11˚08'03.5"E). Three types of samples were taken: environmental, gut and body-surface. At each site, all the samples were collected during the same low tide in July-

August 2018. The snails were collected from a limited coastal area 8–10 meters wide along the shore and 20–25 meters wide along the vertical gradient of the intertidal zone. Since our main aim was to evaluate differences between sympatric *Littorina* spp. taking into account geographical variability, we used pooled samples to reduce the "biological noise" (the effect of between-individual or between-microhabitat variability). In Sweden *L. saxatilis* is represented by the Crab and the Wave ecotype. Only samples of the Crab ecotype from the boulder areas were included in the analysis.

Pooled environmental samples were obtained by multiple scraping of natural substrates (fucoids or boulders) inhabited by periwinkles in different parts of the intertidal zone of the collection site. For the analysis of the associated microbiomes the mid- and the hindgut parts of five randomly selected individuals were pooled within the sample with three replicates (i.e., a total of 15 individuals); the cephalic tentacles of these 15 individuals were pooled into one body surface sample. The samples were fixed with ethanol (70%) (the scraped material immediately at the shore, the snail body parts in the laboratory on the day of collection). The dissection procedure included the breaking of the shell, the inspection of genitalia morphology for species identification (according to [52] and described in detail in previous publications [50, 54]), excision of the tentacles and, finally, resection of the mid- and hindgut parts. Snails infected with trematodes were excluded from the analysis. The species composition differed in the two regions of collection (Table 1).

The Petri dishes used for dissecting the snails were routinely cleaned after each dissection. In order to exclude the possibility of contamination, a negative laboratory control sample was taken by wiping the cleaned Petri dishes with a sterile tampon. The control sample was processed in the same way as the other samples.

## Library preparation and sequencing

DNA was isolated using the DNeasy PowerSoil kit (QIAGEN) following the manufacturer's recommendations. Further workflow included dual indexing 16S rDNA library preparation

**Table 1. Collected samples by sites.**

| **Norway, Tromsø** | | |
|---|---|---|
| | **Tromsø#1 (28 samples)** | **Tromsø#2 (28 samples)** |
| **snail** | *L. fabalis* (gut x3, body surface x1) | *L. fabalis* (gut x3, body surface x1) |
| | *L. obtusata* (gut x3, body surface x1) | *L. obtusata* (gut x3, body surface x1) |
| | *L. littorea* (gut x3, body surface x1) | *L. littorea* (gut x3, body surface x1) |
| | *L. compressa* (gut x3, body surface x1) | *L. saxatilis* (gut x3, body surface x1) |
| | *L. saxatilis* (gut x3, body surface x1) | *L. arcana* (gut x3, body surface x1) |
| | *L. arcana* (gut x3, body surface x1) | *L. compressa* (gut x3, body surface x1) |
| **environment** | *F. vesiculosus* scraping (x1) | *F. vesiculosus* scraping (x1) |
| | *F. serratus* scraping (x1) | *F. serratus* scraping (x1) |
| | *A. nodosum* scraping (x1) | *A. nodosum* scraping (x1) |
| | gravel (x1) | gravel (x1) |
| **Sweden, Tjärnö** | | |
| | **Tjärnö#1 (16 samples)** | **Tjärnö#2 (16 samples)** |
| **snail** | *L. fabalis* (gut x3, body surface x1) | *L. fabalis* (gut x3, body surface x1) |
| | *L. littorea* (gut x3, body surface x1) | *L. littorea* (gut x3, body surface x1) |
| | *L. saxatilis* (gut x3, body surface x1) | *L. saxatilis* (gut x3, body surface x1) |
| **environment** | Boulders scraping (x3) | Boulders scraping (x3) |
| | *F. vesiculosus* scraping (x1) | *F. vesiculosus* scraping (x1) |

for sequencing on the Illumina MiSeq platform by the following steps: inner locus-specific PCR, amplicons clean-up with magnetic beads, outer index PCR, amplicons clean-up with magnetic beads, and final pooling. DNA concentration in the samples at every step was measured using a Qubit 2.0 fluorometer (ThermoFisher Scientific) and the Qubit dsDNA BR Assay Kit (ThermoFisher Scientific, Invitrogen). The library preparation process followed the online Protocol https://github.com/EnvGen/LabProtocols/blob/master/Amplicon_dual_index_prep_EnvGen.rst. The V3-V4 region of the 16S rRNA gene was amplified in Eubacteria using 314F (5'-CCTACGGGNGGCWGCAG -3') and 805R (5'- GACTACHVGGGTATC TAATCC -3') primers from [55]. More details can be found in S1 Text in S1 File. The total of 85 samples was sequenced on the Illumina MiSeq v3 PE 2x300 bp platform by the National Genomic Infrastructure NGI-Sweden, Stockholm.

## Data processing and bioinformatics analysis

The reads were demultiplexed per sample by the National Genomic Infrastructure. Then we assessed the quality of the obtained sequences using the FastQC program (https://www.bioinformatics.babraham.ac.uk/projects/fastqc/). Low-quality nucleotides and adapter sequences were removed g by the *cutadapt* tool [56]. The following steps were carried out using the *mothur* pipeline [57, 58]. Forward and reverse reads were joined into contigs and contigs with ambiguous bases or homopolymer regions, longer than eight nucleotides as well as contigs longer than the expected fragment length were removed. Unique sequences were aligned to SILVA 16S rRNA reference database v132, December 2017 [59, 60], and sequences producing poor alignments as well as chimeric sequences were removed. After that, the sequences were classified at the domain level, and those of eukaryotes, archaea, chloroplasts, mitochondria as well as unclassified sequences were removed. The remaining bacterial 16S rRNA sequences were clustered in operational taxonomic units (OTUs) at 97% similarity level, and taxonomy was assigned using Wang's algorithm within the Mothur software. After that, the sequences were clustered into OTUs (operational taxonomic units). Singletons and sequences with the number of reads at the level of the negative control were excluded from the analysis.

Finally, for OTUs that could not be classified to the genus level by the analysis described above we performed the BLAST search in the NCBI-nt database using the blastn algorithm with default parameters. Microbiome communities were rarefied to 32,729 reads per sample to be included in the analysis.

## Statistics and visualization

The analysis of OTU tables was performed in R [61]. We characterized alpha and beta diversity of the microbiomes. Alpha diversity is a characteristic of taxonomic richness within individual communities [62]. To describe alpha diversity, we assessed taxonomic richness $S$ as the number of OTUs in each sample and calculated the Shannon-Wiener diversity index ($H' = -\sum_{i=1}^{S} p_i log_e p_i$, where $p_i$ is the proportion of individuals of a particular taxon $S$). This index describes the number of taxa and their ratio: the more taxa there are in the community and the less their abundance differs, the higher the value; the stronger the differences in abundance, the lower the value of the coefficient [63, 64]. The Pielou evenness index ($J = H'/log S$) was used to assess uniformity of the abundance distribution among taxa in each sample type: $J_{max} = 1$, $J$ is maximum in the case of an equal species abundance [65].

Beta diversity is a variability of community composition depending on conditions or differences in diversity between communities [62]. To describe beta diversity, one has to characterize the extent of compositional differences in microbiota in each sample type (environmental, mantle and intestinal samples at four sites in two regions). To equalize the effect of OTUs with

different abundances, we pre-standardized the data using double Wisconsin standardization (first, OTU abundances were standardized using their maxima, then the OTU abundances in the samples were divided by the total abundance of all OTUs). Next, the Bray-Curtis dissimilarity matrix was calculated [66]. This matrix was used to ordinate samples according to the similarity of taxonomic composition and OTUs abundance using the non-metric multidimensional scaling (nMDS). The quality of ordination was assessed using the stress value [67]. The analysis was carried out using the *vegan* package [68]. nMDS plots were built using the *ggplot2* package [69].

The effects of the region (North Sea, Norwegian Sea) and sample type (gut, body surface, environmental) on the composition of the whole microbial communities were assessed visually on the nMDS ordination and tested on the dataset of three species sampled in both regions (*L. saxatilis*, *L. fabalis*, *L. littorea*). The permutational analysis of variance (PERMANOVA, [70]) relied on the Bray-Curtis coefficients matrix and was performed in the *vegan* package. The model included region, sample type and the interaction of these factors. Tests were performed on the complete set of permutations (199) restricted within the collection site. PERMANOVA assumptions were fulfilled since no significant differences of intragroup dispersion were found using the *betadisper* function of the *vegan* package.

The effects of the snail species on both the whole gut-associated and the gut-specific microbiome composition were tested separately in the two regions with different sets of species sampled (in the Norwegian sea, *Littorina saxatilis*, *L. arcana*, *L. compressa*, *L. obtusata*, *L. fabalis*, *L. littorea*; in the North Sea, *L. saxatilis*, *L. fabalis*, *L. littorea*). The PERMANOVA was performed on Bray-Curtis dissimilarities between snail microbiomes. The models included two factors: snail species and site. The significance was tested using 10,000 permutations. The conditions of PERMANOVA applicability were met only for gut-specific microbiomes from the North Sea, but not in the three other cases, where *p*-values should be treated with caution due to significant differences in intragroup dispersions. After PERMANOVA, we carried out a series of post hoc pairwise comparisons to assess the significance of differences between the microbiomes. The level of statistical significance was corrected using the Holm-Bonferroni method to keep the type I error probability at the $\alpha = 0.05$ level [71].

The taxonomic composition of the samples and the contribution of major and minor taxa to the total diversity were visualized, reflecting the names of the 20 most abundant OTUs for all samples on bar graphs constructed in the *fantaxtic* package [72].

We used the IndVal index [73, 74] calculated using the *indicspecies* package [75] to identify OTUs specific to certain sample types. The IndVal index evaluates the predictive value of an OTU as an indicator of a particular group of samples. The index consists of two components: *A*–specificity ($0 \leq A \leq 1$)–which is the ratio of the mean abundance of species *i* in site group *j* and the sum of means of the same species over all groups; *B*–fidelity ($0 \leq B \leq 1$)–which is the proportion of sites in which species *i* is present within group *j*. OTUs with a total IndVal value of no less than 0.98 (with A and B values over 0.95) were considered as marker ones. Thus, we identified OTUs found only in the samples of a certain group and in almost all replicates.

The microbiomes specific for each sample type were reflected on heat-trees drawn using the *metacoder* package [76, 77]. We used the heat-trees to visually assess the taxonomic richness of specific prokaryotic microbiota in each sample type at several taxonomic levels.

## Results

### General characteristics of microbiomes

We analysed microbiome composition in the gut and on the body surface of six *Littorina* spp. from Norway and three *Littorina* spp. in Sweden, as well as in environmental samples from the

**Table 2. Total number of OTUs registered in samples of different types.**

| | Total | Specific | *L. saxatilis* | *L. arcana* | *L. compressa* | *L. obtusata* | *L. fabalis* | *L. littorea* |
|---|---|---|---|---|---|---|---|---|
| **Environment_total** | 3404 | 1330 | - | - | - | - | - | - |
| **Environment_mean** | 735 | 187 | - | - | - | - | - | - |
| **Gut_total** | 4766 | 2184 | 2534 | 1247 | 1430 | 957 | 2325 | 1554 |
| **Gut_mean** | 471 | 67 | 555 | 501 | 486 | 388 | 531 | 340 |
| **Body surface_total** | 3042 | 811 | 1380 | 782 | 977 | 351 | 1217 | 1271 |
| **Body surface_mean** | 469 | 54 | 533 | 474 | 603 | 207 | 462 | 473 |
| **Body surface and gut** | 1985 | 754 | 1514 | 998 | 1152 | 752 | 1432 | 1345 |

snail habitats (Table 1). The raw sequence data varied between 65,935–607,687 reads per sample; after filtering, the data volume was 40,890–344,580 reads per sample (S2 Table in S1 File). The raw sequence data are deposited in the NCBI archive, BioProject PRJNA778532. The final dataset included 7,153 OTUs from several types of samples: the environmental samples and the snail-associated samples (from the gut and from the body surface) of the six snail species (Table 2).

The analysis was based on snail populations from two distant regions, Norway and Sweden, with two sites within each region. The OTU composition of the microbiomes varied significantly across sample types and regions. Only three of 7,153 OTUs were recorded in all the samples. They belonged to Proteobacteria: two OTUs of previously undescribed Enterobacteriacea and one of Rhodobacteriacea. In environmental samples, a total of 71 OTUs out of 3,404 were recorded at all four collection sites.

The components of the microbiomes were divided into two categories: those present only in a certain type of samples (specific) and those registered not only in a certain sample type but also in some other types (non-specific). Quite unexpectedly, the maximum total number of OTUs was registered not in environmental samples but in gut ones. Moreover, the gut microbiomes had the highest percentage of specific OTUs (46%), while the smallest percentage of specific OTUs was registered in the microbiomes of the body surface. Interestingly, there was a significant overlap in OTU composition between gut and body-surface microbiomes, greater than that between either of them and the environmental one (Fig 1, Table 2).

Beta-diversity was slightly lower in gut samples than in samples of other types (S3 Fig in S1 File). Accordingly, the percentage of the OTUs shared between gut microbiomes from different sites was higher than that between either body-surface or environmental microbiomes (S4 Fig in S1 File).

## Factors structuring the diversity of microbiomes

The study design allowed us to evaluate the influence of several factors on the microbiome composition: snail species, geographic region and microbiome source (environment or some body part). Multidimensional scaling of whole-microbiome samples showed that the sample type had the strongest effects on microbiome composition. Environmental, gut and body-surface microbiomes were clearly dissimilar in OTU composition (Fig 2A).

Although the effect of the geographic region (Tromsø *vs* Tjärnö) was less obvious, it was still visible on the nMDS plot and confirmed by PERMANOVA ($p = 0.005$). The effect of the molluscan species on the whole gut-associated microbiome composition was also significant (PERMANOVA, $p = 0.002$ in Tromsø, $p < 0.001$ in Tjärnö), though not so pronounced. Many OTUs in the gut- and body-surface-associated microbiomes (or both) were registered for one snail species only (Table 3, Fig 3).

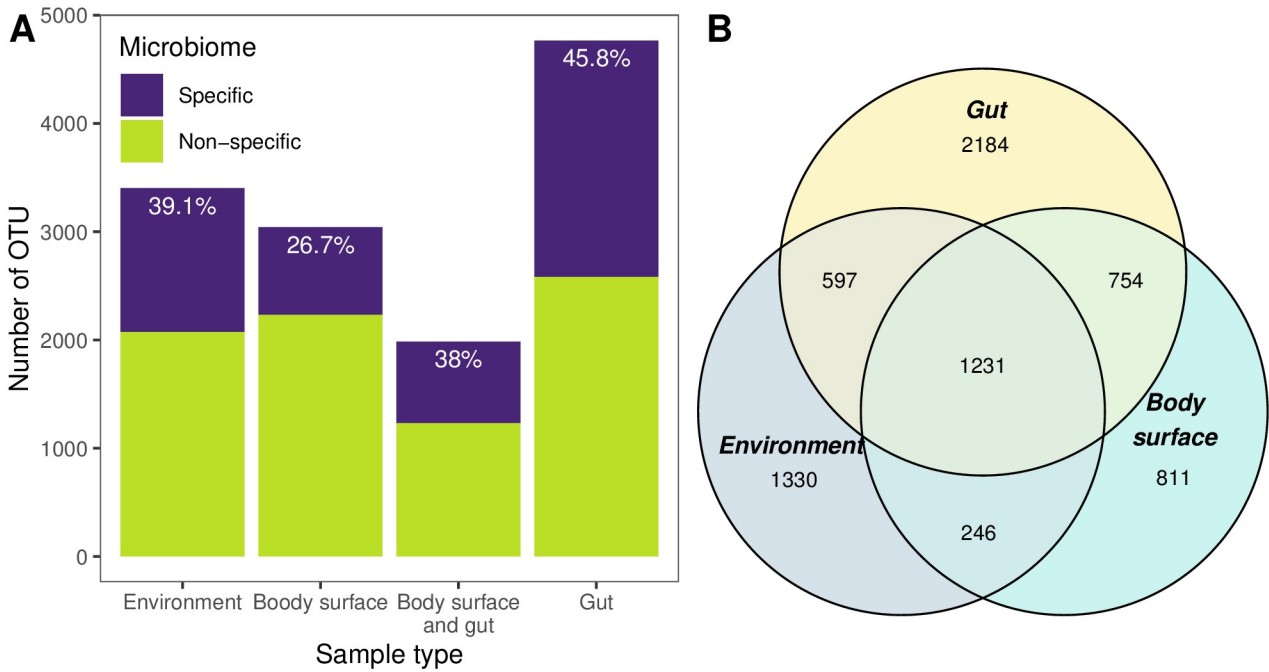

**Fig 1. Taxonomic richness of environmental and *Littorina*-associated microbiomes.** A: The ratio of specific (found in a certain sample type) and non-specific (found in several sample types) OTUs by sample type. B: Venn diagram of the number of OTUs found in different sample types and their combinations. "Environment": OTUs, registered on different types of environmental substrates; "Body surface": OTUs, registered on tentacles; "Body surface and gut": OTUs, registered both on the tentacles and in the gut; "Gut": OTUs, registered in the gut.

Importantly, the differences between *Littorina* spp. became much clearer when only gut-specific (not the whole gut-associated) microbiomes were analysed (Fig 2B). The gut-specific microbiome composition significantly differed between species, and the differences between species were stronger than between sites (PERMANOVA, significant effects of species and site in both regions, p < 0.001). This illustrates that OTUs of the gut-specific microbiome were often species-specific as well, while OTUs registered both in the gut and the environmental (and/or body-surface) samples were usually common across species. Pairwise post hoc comparisons showed significant differences between species of the two species groups ("saxatilis" vs "obtusata"), but not within groups; *L. littorea*, the species of the subgenus *Littorina* (*Littorina*), differed from all *Littorina* (*Neritrema*) spp., with exception of *L. fabalis* in Tromsø populations (S5 Table in S1 File).

Alpha diversity (within-sample taxonomic diversity) was higher in the environmental samples than in the snail-associated ones. This was expressed in mean OTU number per sample (S), Shannon-Wiener index (H) and evenness Pielou index (J) (Fig 4). This implies that associated microbial communities on average had a lower species richness and included a limited number of highly abundant OTUs and numerous rare OTUs. The associated microbiomes of different species had a similar alpha diversity and evenness; microbiomes tended to be less diverse and even in *L. littorea* than in other species, though this tendency was significant only in one pairwise comparison (*L. littorea* vs *L. fabalis*, S6 Table and S7 Fig in S1 File). The plot of relative abundance of the 20 most abundant OTUs confirmed this conclusion (Fig 5).

There were 1–5 strongly dominant OTUs in the snail-associated microbiomes, in contrast to the environmental ones. Noteworthy, these dominant OTUs were conservative across species within a geographic region and, to a lesser extent, across regions, but not across sample

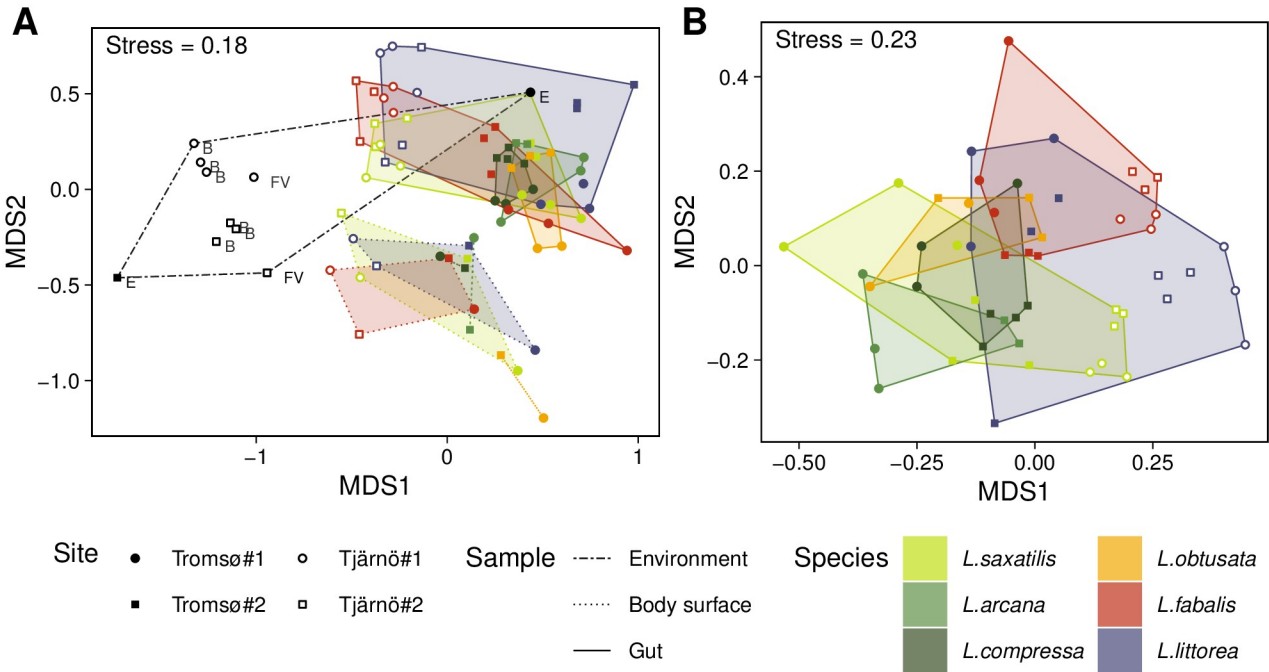

**Fig 2. Comparison of microbiomes.** A: nMDS of the whole microbiomes in different types of samples. B: nMDS of the gut-specific microbiomes of the six *Littorina* spp. C: post-hoc pairwise comparison of microbiomes collected at the Norway Sea coast.

types (these OTUs could be present, though less abundantly, in samples of other types). This indicates that species-specificity and region-specificity of the *Littorina*-associated microbiomes was determined by rare OTUs. The dominant genera were *Vibrio* (γ-Proteobacteria), *Psychromonas* (γ-Proteobacteria), *Psychrilyobacter* (Fusobacteria) and others in the gut microbiome, new species of the order Cytophagales (Bacteroidetes) and the family Enterobacteriacea (α-Proteobacteria) in the body-surface microbiome, etc. The taxonomy of all these OTUs is given in Fig 5.

## Taxonomic composition of microbiomes

The main dominant high-rank taxa were the same in microbiomes from different sources (Fig 5; S8 Fig in S1 File). These were α- and γ-Proteobacteria (Proteobacteria); Bacteroidia (Bacteroidetes were relatively richer in OTUs in environmental microbiomes; the same pattern was observed for Oxyphotobacteria, Cyanobacteria); Planctomycetacea (Planctomycetales, slightly more numerous in the gut microbiomes) and Verrucomicrobia; the same was observed for Mollicutes (Mycoplasmatales), an important, though not dominant group of bacteria. Patescibacteria were relatively more numerous in the environment and on the body surface, where

**Table 3. Number of the species-specific OTU by species and snail body part.**

|  | *L. saxatilis* | *L. arcana* | *L. compressa* | *L. obtusata* | *L. fabalis* | *L. littorea* |
|---|---|---|---|---|---|---|
| **Gut** | 605 | 184 | 164 | 69 | 502 | 234 |
| **Body surface** | 165 | 81 | 154 | 30 | 144 | 153 |
| **Body surface and gut** | 57 | 15 | 8 | 1 | 32 | 22 |

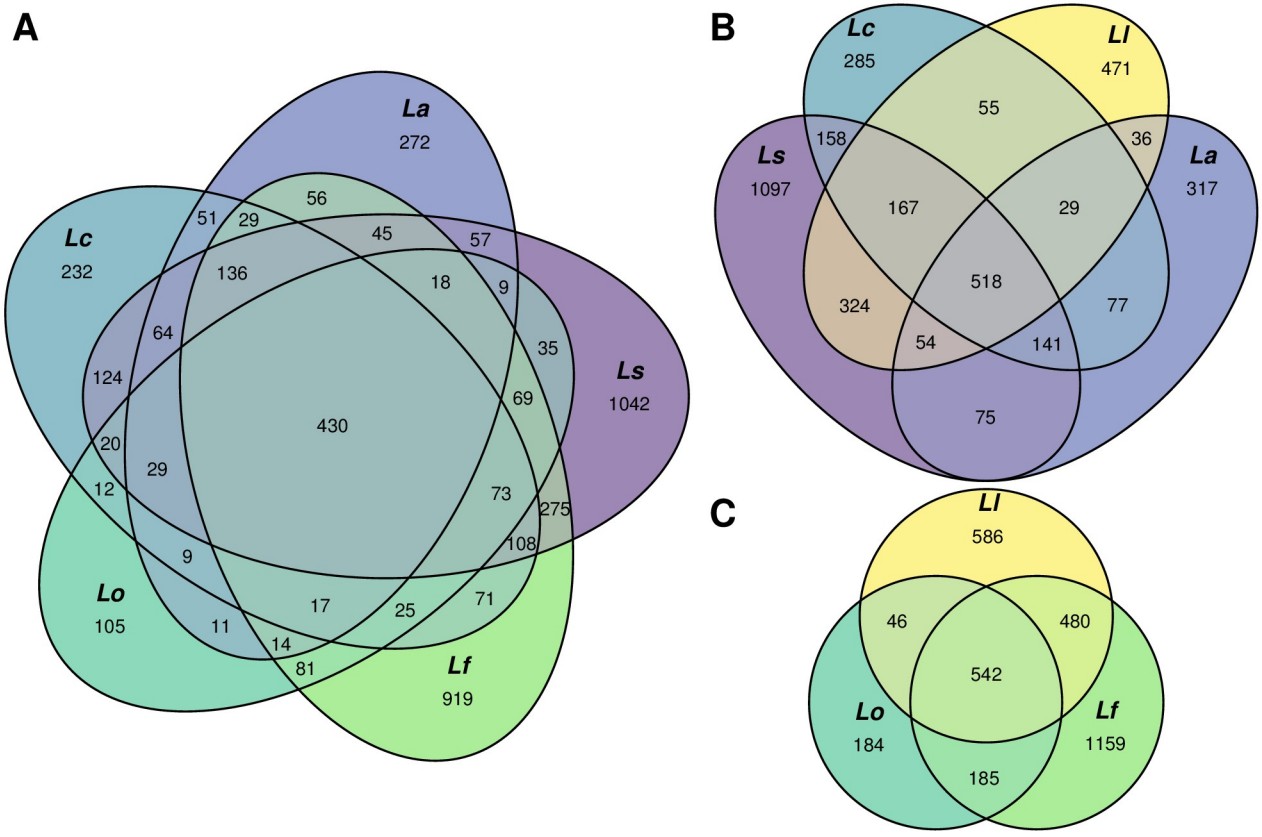

**Fig 3. Venn diagrams of the gut-associated microbiome of different *Littorina* spp.** A: microbiomes of *Littorina (Neritrema)* spp.; B: microbiomes of species of the "saxatilis" group plus *L. littorea*; C: microbiomes of species of the "obtusata" group plus *L. littorea*.

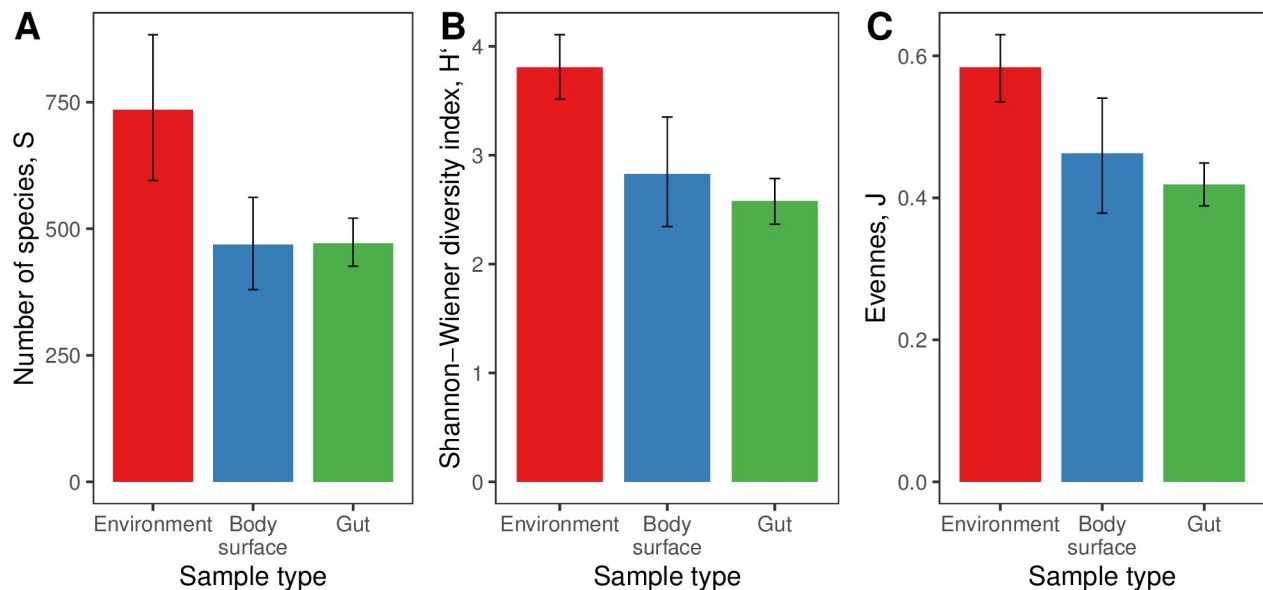

**Fig 4. Alpha diversity of associated microbiomes by sample type.** A: S, taxonomic richness, S (measured as mean OTU number per sample); B: Shannon-Wiener index, H'; C: Peilou's evenness index, J; means with 95% confidence limits obtained via bootstrap with 1000 iterations. The taxonomic richness, Shannon-Wiener and Peilou's indices by sample can be found in S9 Table in S1 File.

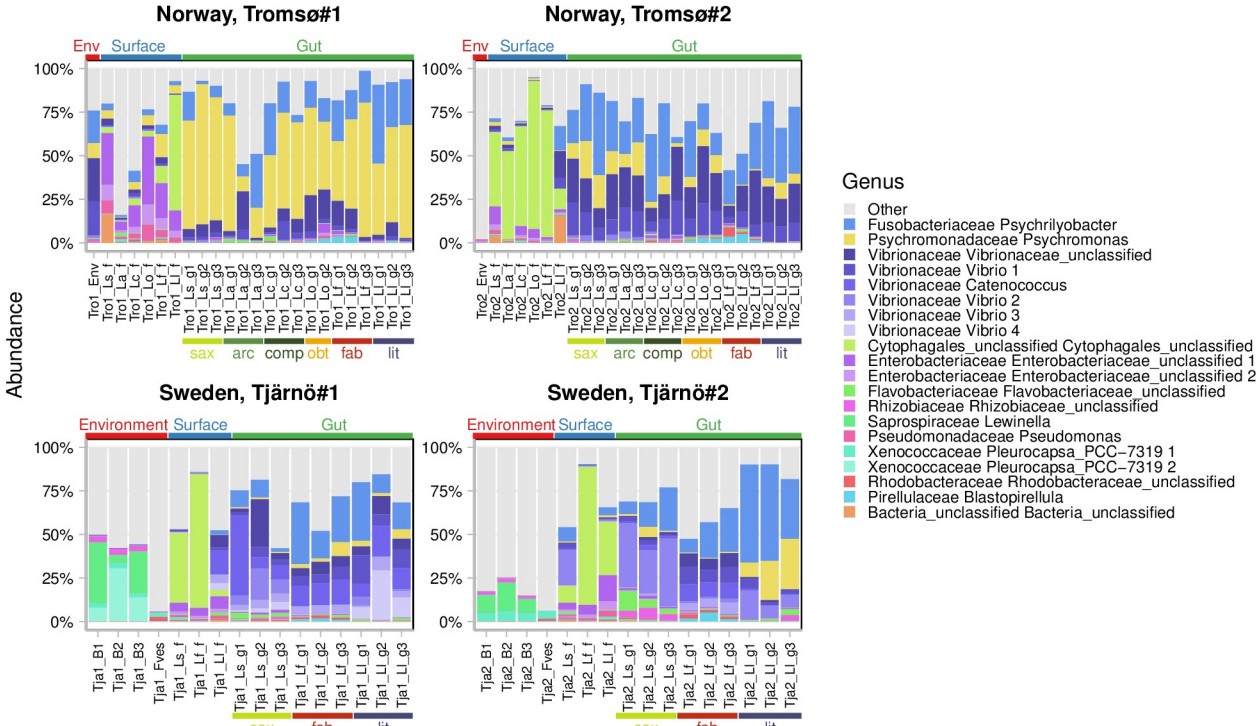

**Fig 5. Relative abundance of OTU at the genus level in environmental and *Littorina*-associated samples.** Colours indicate 20 most abundant OTU in each sample; all other OTUs—are shown in grey. Types of samples are shown above the plot: Env, Environment—environmental samples; Surface—body surface; Gut—gut of snails. Molluscan species are shown below the plot: sax—*Littorina saxatilis*; arc—*L. arcana*; comp—*L. compressa*; obt—*L. obtusata*; fab—*L. fabalis*; lit—*L. littorea*.

they were represented by Gracilibacteria and Parcubacteria, while in the gut they were less diverse and represented mainly by Saccharimonadia.

The relative contribution of Firmicutes Clostridia and Bacilli to taxonomic diversity was higher on the body surface than in the environment and in the gut. Actinobacteria and Acidobacteria were more diverse in the snail-associated microbiomes than in the environment; this tendency was even more prominent for Spirochaetia and Leptospirae (Spirochaeles), as well as Chlamydiae. Noteworthy, the composition of the gut microbiomes was more similar at a higher taxonomic rank than at the level of OTUs (Figs 5 and 6). Taxonomic composition of the specific and the non-specific parts of the gut-associated microbiomes was similar at the level of classes and orders.

OTU abundance in the gut samples was clearly different from that in the body-surface samples and in the environmental ones. While Bacteroidia was often the most abundant taxon in both the body-surface and the environmental microbiomes, γ-Proteobacteria and Fusobacteria evidently dominated in the gut. Oxyphotobacteria were relatively abundant in the environmental microbiomes but rare in the snail-associated ones. Interestingly, Verrucomicrobia were usually more abundant in the samples of *L. fabalis* microbiomes. α-Proteobacteria were approximately equally represented in the microbiomes from different sources (Fig 6).

Individual OTUs dominant in the snail-associated microbiomes were generally the same in different species and often in different regions as well (Fig 5; S4 Fig in S1 File). Some OTUs were registered in all types of samples, e.g., *Psychrilyobacter* sp. or *Psychromonas* sp. profoundly dominated the gut microbiomes (up to 55% and 80%, respectively) and were also

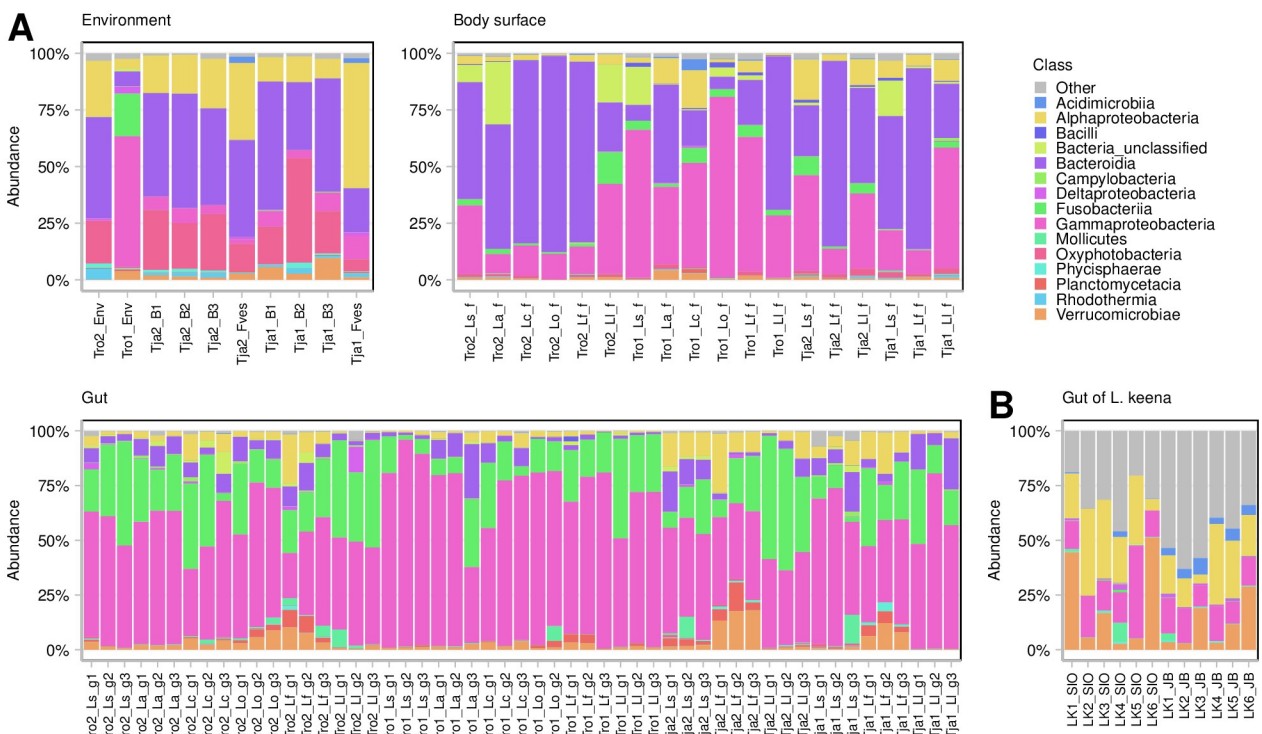

**Fig 6. The distribution of the most abundant high-rank bacterial taxa.** A. Fifteen most abundant bacterial classes in environmental and snail-associated samples (body surface and gut) in this study. B. Abundance of these classes in the gut-associated microbiome of *L. keenae* (data from Neu et al., 2019 kindly provided by the authors).

present in the body surface and the environmental samples, though at much lower levels (no more than 18.8% and 8.6%, respectively). *Vibrio* spp. the most abundant in the environmental samples but were also regularly identified in the gut microbiomes. Unexpectedly, some OTUs, e.g. *Cytophagales* g. sp., *Enterobacteriaceae* g. spp., *Bacteria* g. sp. were dominant in the body surface microbiomes in all the samples (up to 85% of relative abundance in case of *Cytophagales g. sp.*) but were only found at very low frequencies in a few of the gut or the environmental samples. These OTU could not be classified into any known genera.

BLAST analysis of the 16S sequences of these OTUs against the NCBI database returned hits against diverse microorganisms, including marine bacteria and bacteria associated with marine invertebrates. *Psychrilyobacter* sp. showed 99.77% identity to diverse lineages of this genus reported from the Black Sea water column (LR606286.1), polluted sediment of the Bay of Biscay (JQ579948.1), biofilms from St. Andrews Bay (JF825448.1), the gut of *Haliotis discus hannai* etc. Similarly, *Psychromonas* sp. demonstrated 99.57% identity to the environmental lineages of the same genus described from the North Sea (Nature Reserve Kullaberg, KC462950.1, and Helgoland, MW580101.1). The 16S *Cytophagales* g. sp. OTU showed ~92% identity to the uncultured bacteria detected in activated sludge of fish-canning wastewater treatment plants (MW083025.1), on the surface of aquatic angiosperms species (EU542198.1) and in tissues of the coral *Astrangia poculata* (MK176178.1). The OTUs, belonging to *Enterobacteriaceae* g. spp., had maximal identity (99.78%) with several *Enterobacter* lineages, reported from polluted seawater (CU915186.1), natural freshwater (MF439181.1), sewage sludge (MG804176.1), rhizosphere (MT557032.1), human secretions (FM881851.2 and LC378894.1) or cabbage white butterfly midgut (DQ342376.1). The OTUs identified as

*Bacteria* g. sp. showed 93.35–96.51% identity to diverse uncultured bacteria lineages, classified as Bacteria incertae sedis, found in association with the Pacific oyster *Crassostrea gigas* (JF827378.1) or various corals (*Nematostella vectensis*, HQ189553.1; *Lophelia pertusa*, FJ041427.1; *Millepora* sp., HQ288586.1; etc.).

Differences between the gut-associated microbiomes of the *Littorina* spp. were associated with several taxonomic groups of Bacteria (S10 Fig in S1 File). OTUs of Tenericutes (Mollicutes) were typically registered in *L. littorea* samples, and were detected in *L. saxatilis* and *L. obtusata* samples only once. OTUs of Actinobacteria were identified almost exclusively in the species of the "saxatilis" group (their traces were also found in the one *L. littorea* sample). OTUs of Cyanobacteria and Clostridia (Firmicutes) were present in samples of species of the "saxatilis" group and *L. littorea*, but not in samples of species of the "obtusata" group. While diverse Planctomycetacia were abundant in the samples of all species, Planctomycetes OM190 OTUs were registered only in *L. fabalis* only, while several unclassified OTUs of Planctomycetacia were registered only in *L. saxatilis*.

Altogether, fourteen species-specific marker OTUs (not for all species) were identified in particular collection sites (S11 Table in S1 File). All these marker OTUs were identified in the gut, not on the body surface, and were among the rare components of the microbial community. No OTU was found to be associated only with a certain snail species.

There were some unexpected lineages among the marker OTUs. In particular, *Simkania negevensis* was identified as a specific OTU of *L. saxatilis* at the Tjärnö collection site. *Simkania* is a member of Chlamydiales, the order comprising many parasites of metazoans. *S. negevensis* was described as an agent of human pneumonia in Europe, South and North Americas, Israel etc. [78, 79].

## Discussion

In this study we provided the first comparative description of microbial assemblages associated with six *Littorina* spp. Our findings are generally in good agreement with the results of recent studies analysing microbiomes of molluscs ([80–82] etc.), including several gastropods, in particular, *Littorina keenae* [54, 83–90]. Notably, gastropods, in line with other invertebrates (e.g. insects) and in contrast to vertebrates, have a higher proportion of Proteobacteria in their gut-associated microbiomes compared due to Bacteroidetes (dominating in, e.g., mammals) [91]. Although the previously examined gastropods varied in composition, taxonomic richness and community structure of their microbiomes, some general tendencies were revealed, and they were similar to those found in our study. These tendencies are as follows. (1) Snail-associated microbiomes differ significantly from the environmental ones. (2) Microbiomes are highly variable between individuals. Although our analysis was based on pooled samples, significant between-sample variability indicated high degree of that at the interindividual level. (3) Snail-associated microbial communities have an uneven dominance pattern, with a limited number of highly abundant taxa. Common previously reported dominant genera were *Vibrio* (γ-Proteobacteria), *Psychromonas* (γ-Proteobacteria), *Psychrilyobacter* (Fusobacteria) and *Mycoplasma* (Mollicutes). The former three genera were dominant in our snail-associated samples, too, and *Mycoplasma* spp., though not dominant, was also found.

One our unanticipated finding was that the total number of OTUs in the *Littorina* gut microbiomes exceeded that in the environmental samples, though mean alpha-diversity (as measured by the Shannon-Wiener index) was higher in the environmental samples compared to the snail-associated samples. Most probably, some bacterial lineages, while being relatively rare in the environment, are selectively enriched within the gut. A similar phenomenon was reported in studies of sponge microbiomes: more than 90% of the total OTU number was

represented by lineages with a very low abundance (less than 0.01%), and any of these rare OTUs had extremely low abundances in the surrounding water column [92–94] (for more details see S7 Fig in S1 File).

In our study, too, OTUs dominant in the *Littorina*-associated microbiomes (in the gut- or on the body surface) were usually detected in the environment, but their abundance was lower. The apparent compositional differences between the snail-associated and the environmental microbiomes point to the existence of some regulatory mechanisms controlling the structure of the host-associated community. These mechanisms may be determined by the functional properties of the host organism, both active (via immune mechanisms) or passive (via providing a habitat with specified parameters), as well as by competition between bacterial lineages within a host-associated community, as suggested for *Haliotis* snails and other organisms [6, 90, 95]. A higher percentage of the gut-specific OTUs (46%) compared to the percentage of body surface-specific OTUs (27%) in the *Littorina*-associated microbiomes suggests that the control mechanisms in the gut are more stringent. A higher average value of the beta-diversity index of body surface samples compared to gut samples confirms these conclusions.

Our study made it possible to evaluate the differences in the microbiome composition of recently diverged sympatric species. To put it another way, we could assess to what extent an associated microbiome was conserved during ecological speciation, which probably occurred in the Atlantic *Littorina* spp. [42, 48–50, 52]. Recently examined microbiomes of two closely related abalones, *Haliotis fulgens* and *H. corrugata* [90] differed in the OTU composition, which the authors interpreted as a consequence of ecological differences. However, there was a strong overlap at the higher taxonomic level. On the other hand, two allopatric species of the *Elysia* coastal sea slugs demonstrated robust differences between their microbiomes even at the high taxonomic level [85, 87, 96]. These two *Elysia* spp. represent strongly diverged allopatric clades, one from the Atlantic and the other from the Indo-Pacific ecosystem [97].

*Littorina* spp. examined in our study represent varying degrees of evolutionary divergence: very young sister species within the "saxatilis" and the "obtusata" groups clearly separated but still relatively recently diverged (≈ 2 Mya) species of the "obtusata" group vs "saxatilis" group; and *L. littorea*, the species with an older divergence (≈ 15 Mya) [52, 53]. Furthermore, these species were collected from two distant regions (Norwegian Sea vs North Sea). Finally, they have different ecological preferences: within the "saxatilis" group, *L. arcana* inhabits the rocks in the higher intertidal zone, *L. compressa* prefers the lower intertidal zone and is mainly associated with *Fucus vesiculosus*, and *L. saxatilis* partially overlaps with them both but is associated with gravel/boulders/rocks rather than *Fucus* thalli. Two species of the "obtusata" group are associated with different fucoids: *L. obtusata* mainly with *Ascophyllum nodosum* and *F. vesiculosus* (to a lesser extent in the study region), while *L. fabalis* is associated with *F. serratus* [42, 49, 50, 52].

Despite clear ecological differences between the studied *Littorina* spp. (except for *L. arcana* and the upper-shore fraction of *L. saxatilis*), the differences in the gut microbiome composition at the level of individual OTUs were found only between phylogenetically diverged clades (the "saxatilis" group, the "obtusata" group, *L. littorea*), but not between species within the groups. Moreover, the absence of any differences whatever at the high-rank taxonomic level points to a prominent evolutionary conservatism of the microbiome. This result is striking, considering deep functional differences between *L. obtusata* and *L. fabalis* at the proteomic and the metabolomic level [48, 49] as well as the differences in their diet. *L. obtusata*, the only macroherbivore among *Neritrema* spp., feeds on fucoid thalli, being resistant to their toxic phenolic compounds [98, 99]. Other *Neritrema* spp., are believed to be feed on bacterial and microalgal biofilms [52] have also been reported to feed on seaweed [100–102]. The host diet is thought to be the main factor shaping gut microbiota [103]. While more powerful analysis

involving a greater number of replicates might reveal small-scale differences between the gut microbial assemblages within the "saxatilis" and the "obtusata" group, the examination at a higher level (order or class) confirmed the general similarity of the gut-associated microbiome of all *Neritrema* spp. under study.

Our results are consistent with the idea of phylosymbiosis, a correlation between the phylogenetic relatedness of hosts and the similarity of their microbiomes [4]. This phenomenon has recently been reported in several vertebrate and invertebrate taxa [91, 104]. In gastropods, phylosymbiosis has been suggested for two closely related intertidal snails, *Chlorostoma eiseni* and *C. funebralis* [51]. Our findings indicate that phylosymbiosis is also fairly probable in case of *Littorina* spp.

Our analysis would be incomplete without a comparison of our results with those of a recent study, in which the associated microbiome of another *Littorina* sp., *L. keenae*, has been described [51]. *L. keenae*, a Pacific species from the subgenus *Planilittorina*, is deeply diverged from the species examined in our study (~45 Mya). It inhabits rocks and boulders of the upper eulittoral and the littoral fringe zones [52, 53], occupying similar microbiotopes as *L. arcana* and *L. saxatilis* (from the upper intertidal level). However, the microbiomes of these three species are not particularly similar even at a high taxonomic rank (Fig 6). Proteobacteria were the richest and the most abundant taxon in all snail species, but while α- and γ-Proteobacteria were equally abundant in *L. keenae*, the latter clearly dominated the gut-associated microbiome of all Atlantic *Littorina* studied. The contribution of Fusobacteria, Bacteroidetes and Planctomycetes to the gut microbial community was far more substantial in the Atlantic species than in *L. keenae*. Some resemblance between the Atlantic and the Pacific *Littorina* spp. (e.g. high abundance of Proteobacteria, presence of the genera *Vibrio*, *Psychrilyobacter*, *Mycoplasma* etc.) cannot be regarded as particularly strong, which corresponds to a high degree of divergence between these groups.

Altogether, the similarity of the high-rank taxa composition and the dominant genera in all the Atlantic *Littorina* spp. at all the collection sites in this study suggests that high-rank taxa composition of an associated microbiome (a "scaffolding enterotype", which may reflect a functional set required by a host) is more conserved during the host evolution, while an individual lower-rank OTU composition is more variable and determined by the environmental context. In accordance with this idea, dominant OTUs (representing clearly dominating high-rank taxa γ-Proteobacteria and Fusobacteria) were common for the species and the regions, while a highly diverse assemblage of OTUs with a low abundance (groups with no clearly dominating member such as α-Proteobacteria, Bacilli and Verrucomicrobia) varied in different species and strongly differed in different regions.

The dominant inhabitants of the *Littorina* gut in our study were *Psychrilyobacter* (Fusobacteria) and *Vibrio* and *Psychromonas* (γ-Proteobacteria). These genera are known to be associated with marine invertebrates, including gastropods. *Psychrilyobacter* is a strictly anaerobic psychrophilic bacterium inhabiting marine sediments [105] and reported from gut microbiomes of tunicates [106], polychaetes [107], bivalves [82] as well as gastropods with diverse diet types: the rapa whelk *Rapana venosa* (predator, [108]), deep-sea species *Rubyspira osteovora* (whale bone-eating, [89]), cultured species *Haliotis tuberculata* (herbivore, [109]) and *H. discus* [110], the Pacific periwinkle *L. keenae* (micrograzer, [51]), and others [111]. *Psychrilyobacter* can metabolize simple mono- and oligosaccharides (glucose, fructose, N-acetyl-glucosamine, etc.), but not complex algal polysaccharides (starch, cellulose, etc.) to produce acetate and short-chain fatty acids via pyruvate fermentation [105, 109, 112]. *Vibrio* and *Psychromonas* (γ-Proteobacteria) often dominate the gut microbiomes of diverse gastropods, too [84, 89, 90, 108–110]. In contrast to *Psychrilyobacter*, these bacteria can hydrolyse vegetal and algal polysaccharides such as starch, laminarine, alginate, etc. [84, 109]. It has been suggested that

"fermenters" (obligate anaerobes residing in the epithelium-adjacent mucus) metabolically cooperate with "degraders" (facultative anaerobes occupying the gut cavity) to digest algal resources [109]. Given the presence of both these groups in the guts of the studied *Littorina* spp., such cooperation might be expected to function in this case as well.

While dominant OTUs were broadly distributed across collection sites, species-specific OTUs had a low abundance and were usually limited to one collection site. A similar trend was identified after an extensive comparative analysis of sponge microbiomes: species-specific host-bacterial associations involved microbial taxa with a very low abundance [94]. Moreover, this observation is in line with a more general ecological phenomenon: species with a low abundance tend to have a narrow distribution due to their specialized ecological requirements [113, 114]. Nevertheless, these rare species are now recognised to contribute significantly to the community functioning owing to their indispensable traits [115, 116]. Accordingly, the gut-associated low-abundance OTUs are assumed to strongly affect the overall functional characteristics of the microbiome [94, 109]. In case of the *Littorina* snails, the presence of rare species-specific OTUs in the gut microbiomes may reflect the properties of their preferred microbiotopes within the intertidal zone [49].

To conclude, we provided the first detailed description of the microbiome associated with several North Atlantic species of *Littorina* snails. The snail gut microbiome differed from the environmental bacterial community and was enriched with some groups of the microorganisms. The variation of the microbiome composition correlated with the phylogenetic divergence of the host species rather than with their ecological preferences. The microbiome composition was conserved at a high taxonomic level, but could vary at a lower level. Our study lays a solid foundation for further research of the functional role of the microbiome in the evolution of *Littorina* snails.

## Supporting information

**S1 File.**
(PDF)

## Acknowledgments

We thank Kerstin Johannesson for her help with the field sampling at the Swedish sites. We acknowledge the support from the National Genomics Infrastructure in Stockholm funded by Science for Life Laboratory, the Knut and Alice Wallenberg Foundation and the Swedish Research Council. We extend our thanks to SNIC/Uppsala Multidisciplinary Center for Advanced Computational Science for assistance with massive parallel sequencing and access to the UPPMAX computational infrastructure. We are grateful to Natalia V. Lentsman for the thorough review and proofreading of the manuscript.

## Author Contributions

**Conceptualization:** Arina L. Maltseva, Marina A. Z. Panova, Andrei I. Granovitch.

**Data curation:** Elizaveta R. Gafarova, Marina A. Z. Panova.

**Formal analysis:** Marina A. Varfolomeeva, Elizaveta R. Gafarova, Marina A. Z. Panova.

**Funding acquisition:** Arina L. Maltseva, Marina A. Z. Panova, Andrei I. Granovitch.

**Investigation:** Arina L. Maltseva, Elizaveta R. Gafarova, Marina A. Z. Panova, Natalia A. Mikhailova, Andrei I. Granovitch.

**Methodology:** Arina L. Maltseva, Marina A. Z. Panova, Natalia A. Mikhailova.

**Project administration:** Arina L. Maltseva, Marina A. Z. Panova, Andrei I. Granovitch.

**Resources:** Marina A. Varfolomeeva, Andrei I. Granovitch.

**Software:** Marina A. Varfolomeeva.

**Supervision:** Arina L. Maltseva, Marina A. Z. Panova.

**Validation:** Arina L. Maltseva, Elizaveta R. Gafarova.

**Visualization:** Arina L. Maltseva, Marina A. Varfolomeeva.

**Writing – original draft:** Arina L. Maltseva.

**Writing – review & editing:** Marina A. Varfolomeeva, Marina A. Z. Panova, Natalia A. Mikhailova, Andrei I. Granovitch.

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
