## [Decision Letter · Decision Letter 0]

12 Oct 2021

PONE-D-21-25891Divergence together with microbes: a comparative study of the associated microbiomes in the closely related Littorina speciesPLOS ONE

Dear Dr. Panova,

Thank you for submitting your manuscript to PLOS ONE. After careful consideration, we feel that it has merit but does not fully meet PLOS ONE’s publication criteria as it currently stands. Therefore, we invite you to submit a revised version of the manuscript that addresses the points raised during the review process.

We look forward to receiving your revised manuscript.

Kind regards,

Geerat J. Vermeij

Academic Editor

PLOS ONE

Additional Editor Comments (if provided):

Having only one quite positive review in hand, I thought I would not have you wait any longer for a decision. Please do shorten the paper and have a native English speaker edit the text.

Journal Requirements:

2. We note that you are reporting an analysis of a microarray, next-generation sequencing, or deep sequencing data set. PLOS requires that authors comply with field-specific standards for preparation, recording, and deposition of data in repositories appropriate to their field. Please upload these data to a stable, public repository (such as ArrayExpress, Gene Expression Omnibus (GEO), DNA Data Bank of Japan (DDBJ), NCBI GenBank, NCBI Sequence Read Archive, or EMBL Nucleotide Sequence Database (ENA)). In your revised cover letter, please provide the relevant accession numbers that may be used to access these data. For a full list of recommended repositories, see http://journals.plos.org/plosone/s/data-availability#loc-omics or http://journals.plos.org/plosone/s/data-availability#loc-sequencing.

Reviewers' comments:

Reviewer's Responses to Questions

**Comments to the Author**

1. Is the manuscript technically sound, and do the data support the conclusions?

Reviewer #1: Yes

2. Has the statistical analysis been performed appropriately and rigorously? 

Reviewer #1: Yes

3. Have the authors made all data underlying the findings in their manuscript fully available?

Reviewer #1: Yes

4. Is the manuscript presented in an intelligible fashion and written in standard English?

Reviewer #1: Yes

5. Review Comments to the Author

Reviewer #1: This is a nice study and takes the much-studied North Atlantic periwinkles into new and interesting territory. The English is generally of a high standard but I do recommend that you try to get a native English speaker to polish a few rough edges - I have indicated some in sticky notes in a pdf copy of the MS which I shall upload with this report form. A perhaps related point is that the Discussion, at some 33% of the main text, is perhaps a bit long: you could try to shorten it and make the writing more brief and 'tight'.

6. PLOS authors have the option to publish the peer review history of their article (what does this mean?). If published, this will include your full peer review and any attached files.

Reviewer #1: **Yes: **John W. Grahame

---

## [Author Response · Author response to Decision Letter 0]

8 Nov 2021

Dear Editor, 

Thank you very much for assessing our manuscript “Divergence together with microbes: a comparative study of the associated microbiomes in the closely related Littorina species”. We also thank the reviewer, Prof. John W. Grahame, for a positive evaluation and valuable suggestions on the improvement of the manuscript. 

Following the suggestions we asked a colleague to correct the English. Further, we shortened the Discussion from 2,209 to 1,842 words. Hopefully, the manuscript reads much better now. The sequence data have been deposited to the NCBI archive and the accession number is now provided in the manuscript (Line 196). We chose not to deposit the laboratory protocols from the study at protocols.io because we followed an already published online protocol, and the reference is provided in the Methods. 

We hope that with these improvements the manuscript became suitable for publication in the PLoS ONE, and we are looking forward to hear from you at your early convenience! 

Sincerely, 

Marina Panova, 

---

## [Editor Report · Decision Letter 1]

17 Nov 2021

Divergence together with microbes: a comparative study of the associated microbiomes in the closely related Littorina species

PONE-D-21-25891R1

Dear Dr. Panova,

We’re pleased to inform you that your manuscript has been judged scientifically suitable for publication and will be formally accepted for publication once it meets all outstanding technical requirements.

Kind regards,

Geerat J. Vermeij

Academic Editor

PLOS ONE
---

## [Editor Report · Acceptance letter]

23 Nov 2021

PONE-D-21-25891R1 

Divergence together with microbes: a comparative study of the associated microbiomes in the closely related *Littorina* species 

Dear Dr. Panova:

I'm pleased to inform you that your manuscript has been deemed suitable for publication in PLOS ONE. Congratulations! Your manuscript is now with our production department. 

Kind regards, 

on behalf of

Dr. Geerat J. Vermeij 

Academic Editor

PLOS ONE